# Effectiveness of Intermediate Respiratory Care Units as an Alternative to Intensive Care Units during the COVID-19 Pandemic in Catalonia

**DOI:** 10.3390/ijerph19106034

**Published:** 2022-05-16

**Authors:** Marina Galdeano Lozano, Julio César Alfaro Álvarez, Núria Parra Macías, Rosario Salas Campos, Sarah Heili Frades, Josep Maria Montserrat, Antoni Rosell Gratacós, Jorge Abad Capa, Olga Parra Ordaz, Francesc López Seguí

**Affiliations:** 1Unidad de Ventilación y Cuidados Respiratorios Intermedios, Servicio de Neumología, Direcció Clínica Àrea del Tórax, Hospital Universitari Germans Trias i Pujol, IGTP, Universitat Autónoma de Barcelona, Reseau Europén de Recherche en Ventilation Artificielle (REVA), 08193 Barcelona, Spain; 2Unidad de Economía de la Salud, Dirección de Innovación de la Gerencia Territorial Metropolitana Norte, Institut Català de la Salut, 08007 Barcelona, Spain; flopezse.germanstrias@gencat.cat; 3Doctorat de Medicina i Recerca Translacional, Facultat de Medicina, Universitat de Barcelona, 08193 Barcelona, Spain; oparra@ub.edu; 4Servicio de Neumología, Hospital de Viladecans, 08840 Barcelona, Spain; jucealal@yahoo.es; 5Unidad de Innovación Clínica y Promoción de la Salud, Hospital Universitario Sagrat Cor, Grupo Quirón Salud, 08029 Barcelona, Spain; nuriaparramacias@gmail.com; 6Servicio de Medicina Interna, Hospital Universitario Sagrat Cor, Quirón Salud, 08029 Barcelona, Spain; rsalascampos@yahoo.es; 7Unidad de Cuidados Intermedios Respiratorios, Hospital Fundación Jimenez Díaz, Grupo Quirón Salud, Reseau Europén de Recherche en Ventilation Artificielle, 28040 Madrid, Spain; sheili@fjd.es; 8Unidad del Sueño, Servicio de Neumología, Hospital Clínic Provincial Barcelona, Universitat de Barcelona, 08193 Barcelona, Spain; jcanal@clinic.cat; 9Servicio de Neumología, Direcció Clínica Àrea del Tórax, Hospital Universitari Germans Trias i Pujol, IGTP, Universitat Autónoma de Barcelona, 08193 Barcelona, Spain; arosellg.germanstrias@gencat.cat (A.R.G.); jabadc.germanstrias@gencat.cat (J.A.C.); 10Unidad de Sueño Servicio de Neumología, Hospital Universitario Sagrat Cor, Quirón Salud, 08029 Barcelona, Spain

**Keywords:** respiratory care, COVID, ICU, cost-effectiveness

## Abstract

**Objectives:** During the COVID-19 pandemic, the risk of collapse for the health system created great difficulties. We will demonstrate that intermediate respiratory care units (IRCU) provide adequate management of patients with non-invasive respiratory support, which is particularly important for patients with SARS-CoV-2 pneumonia. **Methods:** A prospective observational study of patients with COVID-19 admitted to the ICU of a tertiary hospital. Sociodemographic data, comorbidities, pharmacological, respiratory support, laboratory and blood gas variables were collected. The overall cost of the unit was subsequently analyzed. **Results:** 991 patients were admitted, 56 to the IRCU (from a of 81 admitted to the critical care unit). Mean age was 65 years (SD 12.8), Barthel index 75 (SD 8.3), Charlson comorbidity index 3.1 (SD 2.2), HTN 27%, COPD 89% and obesity 24%. A significant relationship (*p* < 0.05) with higher mortality was noted for the following parameters: fever greater than or equal to 39 °C [OR 5.6; 95% CI (1.2–2.7); *p* = 0.020], protocolized pharmacological treatment [OR 0.3; 95% CI (0.1–0.9); *p* = 0.023] and IOI [OR 3.7; 95% CI (1.1–12.3); *p* = 0.025]. NIMV had less of a negative impact [OR 1.8; 95% CI (0.4–8.4); *p* = 0.423] than IOI. The total cost of the IRCU amounted to €66,233. The cost per day of stay in the IRCU was €164 per patient. The total cost avoided was €214,865. **Conclusions:** The pandemic has highlighted the importance of IRCUs in facilitating the management of a high patient volume. The treatment carried out in IRCUs is effective and efficient, reducing both admissions to and stays in the ICU.

## 1. Introduction

In March 2020, the SARS-CoV-2 [1] pandemic was declared in Spain. From 25 February to 28 April 2020, a total of 203,715 cases were confirmed, of which 105,743 were older than 60 years; of these, 15,115 died, and 4353 (4.12%) required admission to an intensive care unit (ICU) [2,3,4,5,6].

The number of ICU beds in Spain is lower than in neighboring countries (approximately 3600, equivalent to a ratio of 7.7 beds per 100,000 inhabitants, compared to 29.2/100,000 in Germany) [7,8,9,10], and the pandemic exposed the scarcity of this resource. The risk of the health system collapsing, especially during the initial phase of the pandemic, created major difficulties for the allocation of care resources and sparked a broad ethical debate.

Although the majority of people with COVID-19 develop mild symptoms or few complications, approximately 14% develop a serious illness that requires hospitalization, and 5% require admission to an ICU [9]. To facilitate difficult decision-making in ICUs during the COVID-19 pandemic, an ethical consensus was created [11,12]. This contained profiles for admission and location priorities in which semicritical units and/or intermediate respiratory care units (IRCUs) played a leading role.

An IRCU is defined as an area for monitoring and providing assistance to patients with acute respiratory failure who require non-invasive mechanical ventilation (NIMV) [13] and/or high-flow oxygen therapy (HFOT) as part of their treatment. Therefore, IRCUs can be a good alternative to an ICU for the treatment of patients admitted for COVID-19 [14,15,16,17] who do not require imminent orotracheal intubation (IOI) or are experiencing clinical improvement.

The implementation of IRCUs is not yet universal in our health system, with a consequent increase in health care expenditure and limitations in the use of adequate resources in each case [18,19,20,21]. IRCUs allow for an adequate selection of subsidiary patients who would benefit from invasive mechanical ventilation (IMV) [14,15], while also treating patients with a more advanced age and/or with associated frailty. This characteristic has become more evident in patients with SARS-CoV-2 pneumonia in need of respiratory support [16,17]. Considerable [18,19,20,21] economic savings have been described for these units.

In this context, our study aims to describe the efficiency of an IRCU at a tertiary hospital, as well as the epidemiological and clinical characteristics and mortality of patients hospitalized in said unit.

## 2. Methods

### 2.1. Definitions

This is a prospective observational study of patients with COVID-19 admitted to the IRCU of a tertiary university hospital during the first wave of the pandemic (from 25 February to 28 April 2020). Ethical approval was obtained from the Ethics and Research Committee.

The University Hospital in Barcelona has 295 general ward beds, 10 ICU beds and six IRCU beds. The IRCU is part of the Pulmonology Service and specializes in non-invasive respiratory support (NIRS) and postsurgical recovery from thoracic surgery. This unit has a closed structure with individual rooms; in contrast, the ICU has an open structure. The IRCU’s nurse-to-patient ratio is 1:4 and the main admission criterion is respiratory failure requiring NIRS (when IMV is needed, patients are transferred to the ICU). The IRCU is equipped with a maximum of six individual bays and continuous non-invasive monitoring, ventilators for NIMV (telemetered) and HFOT. The respirators available are V60 and Trilogy 200 Phillips Respironics^®^. During the first wave of the pandemic, a maximum of 11 bays were made available with these devices.

In this period, the manner in which personnel were assigned was modified. Day care was 12 h long with pulmonologists in charge. Continuous night care was provided by intensive care physicians at the center. Staffing in terms of nurses and assistants with experience in non-invasive respiratory support techniques consisted of two nursing graduates and two nursing care technicians. The unit received support from the rehabilitation service in the form of physiotherapy.

### 2.2. Inclusion and Exclusion Criteria

All cases of SARS-CoV-2 pneumonia confirmed by real time polymerase chain reaction (RT-PCR) for SARS-CoV-2 admitted to the IRCU were consecutively included. Admission criteria to the IRCU were as follows: a ratio of arterial oxygen pressure and fraction inspired oxygen (PaFiO2) lower than 50% and oxygen saturation lower than 95%, or respiratory frequency (RR) higher than 25 rpm and PaFiO_2_ lower than 250.

This group of patients included all patients with NIRS needs (experiencing clinical deterioration and possibly requiring admission to the ICU, those making a recovery, discharges from the ICU, and patients who had reached a therapeutic ceiling). We consider that not all dying patients have ICU admission criteria (given age and/or associated multiple pathologies), so we present the analysis described below.

### 2.3. Statistical Analysis

Sociodemographic variables (age, gender), clinical variables (Barthel index, Charlson comorbidity index) and chronic and acute comorbidity, obesity and arterial hypertension were collected in addition to home respiratory therapeutic requirements prior to admission and at the time of discharge (NIMV, continuous positive airway pressure (CPAP) and home oxygen therapy); variables of the care process (average length of stay and hospital stay, diagnosis at admission, discharge status), as well as analytical and microbiological parameters, treatments during admission, destination on admission (ICU, IRCU or general ward) and support needs during their stay (NIMV, IMV and IOI, HFOT or CPAP). The costs derived per day of stay were estimated and compared with the costs of an ICU stay. To evaluate the efficiency of the unit, the days of admission to the ICU avoided were considered for patients assessed and taken on by the IRCU at the time of requiring NIRS, as well as post-ICU patients who were discharged early from the unit. The number of stays for patients admitted to the unit was calculated and the theoretical bed cost estimated for the ICU was calculated according to Section 1 of the Spanish Ministry of Health’s classification [20]. The difference between the real cost calculated for the IRCU and the estimated cost for the ICU was considered to be the cost avoided.

Results are expressed as the mean and SD for quantitative variables that follow a normal distribution and as the median and IQR otherwise. The variables were described and compared according to their nature and distribution. Qualitative variables are expressed as total number and percentage. With respect to the mortality predictors, a univariate analysis was included in the corresponding multivariable logistic regression backward stepwise model. Strongly correlated variables were excluded from the analyses. All tests were performed with a bilateral significance level of *p* = 0.05. Statistical analysis was performed using SPSS statistical software (15th version).

## 3. Results

A total of 991 patients confirmed with COVID-19 were admitted during the study period. Of these, 910 were admitted to a general hospital ward and 81 were admitted to a critical care unit. A total of 32 patients (39.5%) were admitted directly to the IRCU and 25 patients (30.8%) were admitted to the UCI.

After an average period of 3.5 days (SD 1.8) of hospital stay, 16 patients (66.7%) worsened and required admission to the IRCU. Of the 81 patients admitted to a critical care unit, 56 were admitted to the IRCU (69.13%). Patient flow is outlined in Figure 1.

The patients admitted to the IRCU had a mean age of 65 years (SD 12.8), a Barthel Index of 75 (SD 8.3), a Charlson comorbidity index of 3.1 (SD 2.2) and the following comorbidities: arterial hypertension accounted for 27%, chronic obstructive pulmonary disease (COPD) was found in 89% of patients, and obesity was found in 24% of patients.

Treatment was sought due to cough and/or dyspnea and/or fever greater than or equal to 39 °C in 65.3% of the cases with an evolution of 7.1 days (SD 4.0), whereas 76.4% presented with bilateral infiltrates according to a chest X-ray performed upon admission.

Upon admission to the IRCU, patients had a sequential organ failure assessment (SOFA) score of 4.5 (SD 2.2), a RR of 23.1 rpm (SD 5.2) and PaFiO2 of 136.2 (SD 65.1). For the initial arterial blood gas analysis, we obtained mean values upon admission of pH 7.43 (SD 0.09), PaO2 67.5 mmHg (SD 37.5) and PaCO2 39.7 mmHg (SD 9.1) (Table 1).

The description of pharmacological treatment was governed by the protocol in force at our center at all times. A total of 68.1% of patients received hydroxychloroquine (100 mg/day), lopinavir/ritonavir (200/50 mg/day) and azithromycin (500 mg/day). In 9.7% of the cases, boluses of corticosteroids (250 mg/day) were administered, 44.4% were administered corticosteroids at 1 mg/kg of weight/day and 6.9% were administered tocilizumab (600 mg in a single dose). A total of 41.7% of the patients received anticoagulation therapy (1 mg/kg/12 h of enoxaparin or equivalent) due to a high clinical suspicion of a thromboembolic event.

The NIRS that patients received upon admission to the IRCU consisted of oxygen therapy through a nasal cannula and/or Venturi mask (69 patients, 85.2% of the total), NIMV (11 patients, 13.6%), CPAP (2 patients, 2.5%), high-flow nasal cannulas (HFNC) (24 patients, 29.6%) and IOI (39 patients, 48.1%). Prone positioning was performed on 48 patients (59.3% of the total). Patients on a Venturi mask received an average flow of 12.13 liters per minute (lpm) with an SD of 6.7 lpm, whereas those on a HFNC received an average flow of 30.96 lpm (SD 12.6 lpm) and 69.50% FIO2 (SD 19.8%). Four simultaneous HFNCs were the only option available.

In those patients who received support with NIMV, the mean inspiratory positive airway pressure (IPAP) was 17.64 +/− 3.91 cmH_2_O, reaching a maximum of 25 cm H_2_O, and the mean expiratory positive airway pressure (EPAP)was 9.27 +/− 2.24 cm H_2_O. NIMVs were used for an average of 4.45 days (SD 4.76 days). Patients who needed more days of NIRS use experienced nasal bridge ulcers and some cases of epistaxis.

The mortality rate of the 81 patients admitted to the critical care unit was 25%. PaFiO_2_ at admission, PaO_2_, heart rate, RR, levels of lactate dehydrogenase (LDH), potassium, alanine aminotransferase (ALT) and total bilirubin were all significantly associated with higher mortality (Table 2). We did not obtain sufficient sputum cultures to describe the most observed respiratory superinfection.

A significant relationship (*p* < 0.05) was found for higher mortality in patients with fever greater than or equal to 39 °C [OR 5.6; 95% CI (1.2–2.7); *p* = 0.020] at the time of admission, with the use of protocolized pharmacological treatment [OR 0.3; 95% CI (0.1–0.9); *p* = 0.023] (hydroxychloroquine, lopinavir/ritonavir and azithromycin) and IOI [OR 3.7; 95% CI (1.1–12.3); *p* = 0.025]. Treatment with intravenous corticosteroids was close to statistical significance for the reduction of mortality [OR 0.3; 95% CI (0.1–1.1); *p* = 0.0508]. The use of NIMV was not statistically significance and it had a smaller negative impact [OR 1.8; 95% CI (0.4–8.4); *p* = 0.423] than IOI (Figure 2).

Description of the cost avoided was based on the total cost of the unit. A calculation of the hospitalization costs of the IRCU was performed for a total of 56 patients over a period of two months based on the rates detailed in a study carried out at our center prior to the COVID-19 pandemic [21].

The total number of stays amounted to 403.2 days with an average of 7.2 days per admission (range 3.8–11 days) and the following itemized expenses:

Personnel costs, taking into account the totality of the existing workforce during all work shifts and the current salary agreement: €52,020.

Expenses for the consumption of sanitary material, instruments, clothing, and linen, among others: €5272.

Amortization and equipment: Most of the material used in the unit had been purchased more than four years earlier, and the remaining material was on loan. No new equipment had been purchased. The equipment inventoried in the unit (not on loan) had a purchase cost of €17,795. None of these expenses are reflected in the period of time that was under study.

Testing and radiology: This cost was calculated based on current care protocols rather than detailed costs for each patient. Three tests were performed as per protocol during a patient’s stay (upon admission, at 48 h and at 6–7 days); X-rays were performed upon admission and at 48 h. Red blood cell and platelet transfusions were taken into account. Therefore, the cost amounted to €3816.

Pharmacy: An exhaustive comparison was carried out between the drugs prescribed in the IRCU and those prescribed in the ICU during the period studied. A dispensing cost was obtained per patient during their stay in the IRCU according to the price of each product, and a comparison of expenses with respect to the ICU was performed. In the IRCU, the average pharmaceutical cost per patient and stay was €91.51 (SD €221.73) compared to €163.83 in the ICU (SD €150). Figure 3 shows that average expenditure in the IRCU was higher than the ICU; nevertheless, since fewer patients were admitted to the IRCU, the figure is based on a smaller dataset. There was no significant difference between the average cost of dispensing a drug in the ICU and in the IRCU.

The total pharmacy expenditure amounted to €5124 (some €3713 more than the previous total of €8468 during the pre-pandemic period according to a recent study [21]. The total cost of the IRCU during the study period as the sum of the categories described above amounted to €66,233. When applied to 403.2 days of stay during this period, the cost per day for the IRCU amounted to €164 per day per patient.

The IRCU saved the hospital a cost equivalent to 403.2 patient days with a theoretical value of €281,098 according to Section 1 of the Spanish Ministry of Health’s classification for the ICU. The difference between the actual cost calculated for the IRCU and the estimated cost for the ICU was considered to be the cost avoided. Therefore, after removing the cost for the IRCU, the actual cost avoided during these two months of the pandemic was €214,865 for 56 patients (Table 3).

## 4. Discussion

During the pandemic, the number of ICU beds was insufficient given the increase in demand, a situation for which the health system was not fully prepared. Several studies and protocols have outlined possible scenarios where it would be necessary to increase the number of ICU [18,19] beds. However, little was published during the first wave of the pandemic with regard to NIRS and the need for an intermediary step before reaching the ICU. The IRCU has been working to avoid IOI and/or to assist in early extubations.

During the period analyzed, the IRCU was essential for the organization of both our patients and the hospital. Note that 40% of our patients with COVID-19 were admitted directly to the IRCU. This percentage is higher than those who were initially admitted to a general ward or the ICU, and this figure increases to practically 70% of those infected by SARS-CoV-2 when we assess the total number of patients who required IRCU services at any time during their hospitalization. In this way, the IRCU offered support to the ICU (and the general ward) at a time when admission criteria played a fundamental role.

Heili-Frades et al. [20] states that this type of unit represents a very important economic saving in terms of avoided expenses. An IRCU can treat patients of a high complexity whose main axis of treatment is NIMV, IMV by tracheostomy or HFOT. The savings occur because this type of unit avoids prolonged or unnecessary stays in the ICU and provides support to those patients whose therapeutic ceiling is NIRS. If these resources had not been available in the IRCU (due to bed shortages), some of these patients would have been treated in a general ward.

In our study, we performed an analysis of the overall consumption of the unit. The unit’s pharmacy cost also allowed us to make a comparison with that of the ICU, which was lower overall in the IRCU than in the ICU. We reached the conclusion that the cost of an IRCU bed in our center during the period under study amounted to slightly less than €200 per day. The fact that pharmaceutical expenditure was higher in the IRCU than the ICU can be attributed to our internal protocol and following current scientific evidence. In the case of NIRS, drugs that have evidence for use are not available to the patient under IoT. However, the concept of avoided cost should not be considered solely from an economic point of view; the added value of a potential improvement in ICU admission capacity should also be considered together with its benefits for the organization of highly complex medical and surgical activities.

The values associated with a worse prognosis were mainly onset with high fever and greater gasometric involvement, regardless of age, gender, previous functional status or the presence of comorbidities. A beneficial trend for treatment with corticosteroids (1 mg/kg) and a worse prognosis associated with IOI was observed. NIMV could be proposed as an alternative treatment in some cases, since the use of non-invasive mechanical ventilation, which was not statistically significant, had less negative impact than IOI.

Invasive measures such as IMV provide benefits but also drawbacks or deleterious effects. It is essential to determine the exact allocation of patients and avoid ICU stays for those who do not benefit from their stay [14,15]. Up to 40% of patients admitted to an ICU do not require intubation, and only 40% of cases of acute respiratory failure require IMV [13], which highlights issues of inadequate resource management.

Reports on hospitalizations due to SARS-CoV-2 pneumonia usually refer to admissions to the ICU, whereas the number admitted to the IRCU is often ignored. Nevertheless, these units have been shown to be effective and efficient given their high degree of specialization. This study looks at the key role played by IRCUs during the start of the pandemic in Catalonia.

In conclusion, the COVID-19 pandemic has highlighted the importance of this type of unit in hospitals since they facilitate the management of a high volume of patients with severe respiratory failure and high dependence. In our study, the treatment performed was effective and efficient, reducing both admissions and stays in the ICU.

## Figures and Tables

**Figure 1 ijerph-19-06034-f001:**
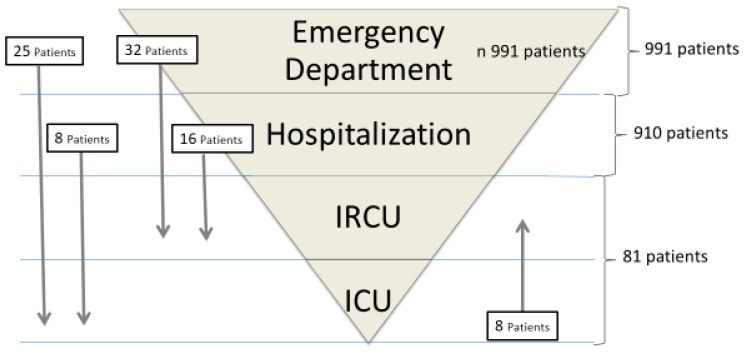
Patient flow.

**Figure 2 ijerph-19-06034-f002:**
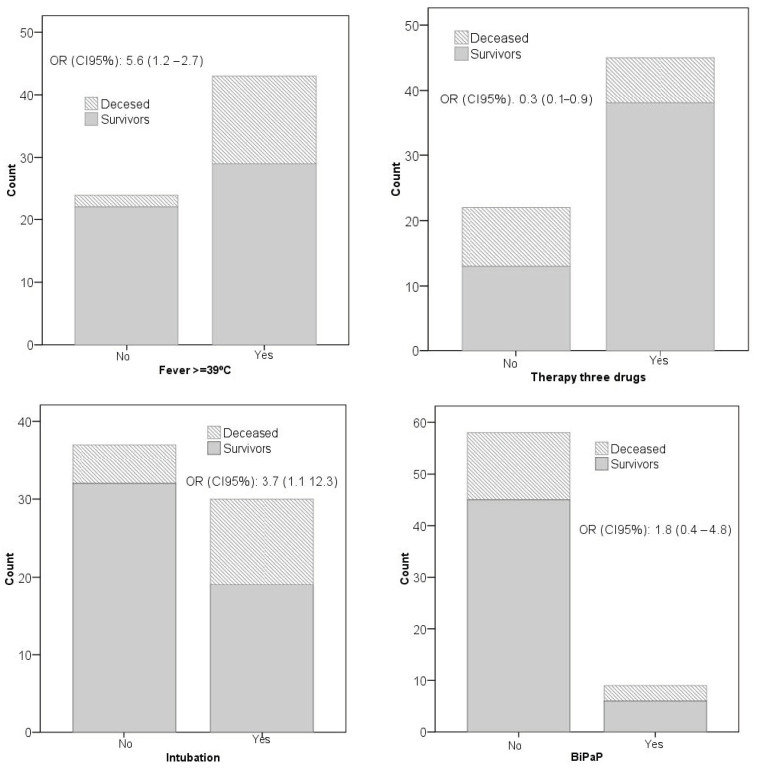
Representation of the significant associations of qualitative variables with mortality.

**Figure 3 ijerph-19-06034-f003:**
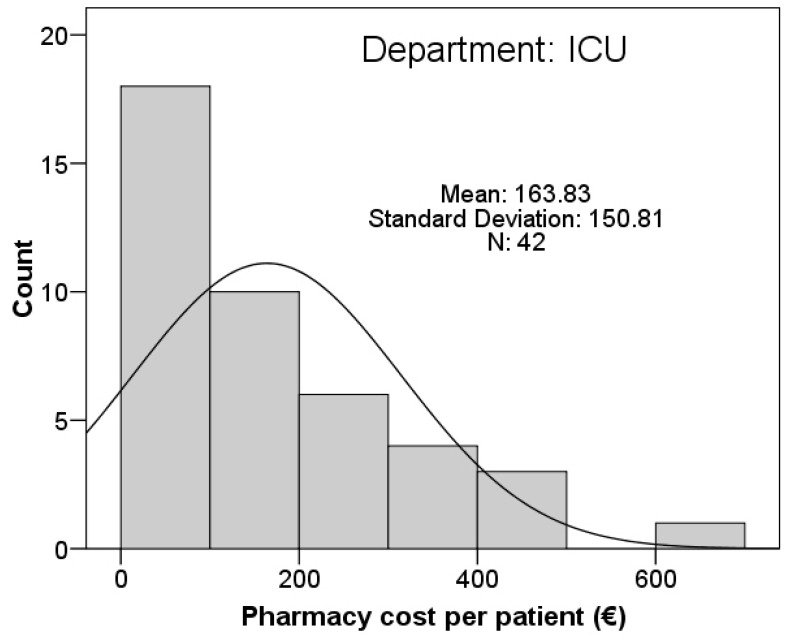
Comparison of pharmacological dispensing cost and number of dispensations per patient between the intermediate respiratory care unit (IRCU) and the intensive care unit (ICU).

**Table 1 ijerph-19-06034-t001:** Characteristics of patients in IRCU, including SOFA (sequential organ failure assessment), Charlson comorbidity index and Barthel comorbidity index.

Clinical Variables	Mean	SD
*Age*	65	12.8
*Barthel Score*	75	8.3
*Charlson Score*	3.1	2.2
*SOFA Score*	4.5	2.2
**Arterial Gasometry Variables**	**Admitted to IRCU** **(Mean/SD)**	**3 Days after Admission to IRCU** **(Mean/SD)**
*pH*	7.43/0.09	7.43/0.13
*pO_2_ mmHg*	67.5/37.5	77.5/34.4
*pCO_2_ mmHg*	39.7/9.1	42.8/9.5

**Table 2 ijerph-19-06034-t002:** Clinical variables and tests associated with mortality.

Test Variables	Deceased	Survivors	*p*
* LDH (mean [SD]) *	649 UI/L [649.7]	327 UI/L [327.0]	0.002
* K+ (mean [SD]) *	3.8 mEq/L [0.58]	4.5 mEq/L [0.65]	0.025
* ALT (mean [SD]) *	81.8 UI/L [63.0]	38.6 UI/L [15.47]	0.004
* Total bilirubin (mean [SD]) *	1.3 mg/dL [0.79]	0.6 mg/dL [0.48]	0.009
Arterial bod gas analyis	Deceased	Survivors	* p *
* PaO2 (mean [SD]) *	44.5 mmHg [25.3]	78.9 mmHg [34.4]	0.034
* PaFiO2 (mean [SD]) *	89.7 [26.3]	147.2 [66.0]	0.002
Clinical Variables	Deceased	Survivors	* p *
* HR (mean [SD]) *	97.5 lpm [20.2]	84.2 lpm [17.187]	0.015
* RR (mean [SD]) *	28.9 rpm [4.9]	24.4 rpm [6.1]	0.013

**Table 3 ijerph-19-06034-t003:** Hospitalization costs in the IRCU for a total of 56 patients over a two month period during the pandemic.

Variables	Cost (€)	% Respect Total Cost the IRCU
Personnel costs	52,020.95	78.54%
Expenses for consumption	5272.27	7.96%
Amortization and equipment	0	0
Testing and radiology	3816.00	5.76%
Pharmacy	5124.56	7.73%
Total cost of the IRCU	66,233.78	100%
Total cost avoided by the IRCU during two months of the pandemic for the 56 patients	214,865.16	

## Data Availability

The results presented are stored in the database of Hospital Universitario Sagrat Cor.

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
