# Peer review of "Effectiveness of Intermediate Respiratory Care Units as an Alternative to Intensive Care Units during the COVID-19 Pandemic in Catalonia"

_ijerph, 2022, doi:10.3390/ijerph19106034_

Round 1

Reviewer 1 Report

This article assess the importance of intermediate respiratory care units (IRCU) as a support to ICU, in the context of the initial phase of the COVID-19 pandemic, when allocating care resources presented major difficulties.

This article presents IRCU as an alternative to alleviate high number of patients in ICU, and seemed to be associated with an overall reduced cost.

Although the objectives of the study are clearly defined and discussed throughout the study, I have some comments regarding data that are missing and other that could be better presented.

The text cites 2 tables (Table 1 and table 2) in the result section but they are not included while they could provide crucial additions.

'he NIRS which patients received upon admission to the IRCU': a typo is to be fixed in this sentence.

Figure 2: could gain in clarity and it is difficult to understand. I suggest changing the fill colors or add a pattern to differentiate the survivors / deceased which appears the same color in black and white.  The 4 panes in this figure don't have a Y axis. I suggest adding 4 letters (A-D), one for each pane, and to indicate in the figure legend (A= fever; B= Therapy three grugs ... ) with additional details.

Figure 3:  this figure is really poor quality and none of the text on both X and y axes is readable as well as the text on the right of the curves.

Author Response

Dear Reviewer,

Below you will find our answers to the questions you have raised. Thank you very much for your time in reviewing the text.

Yours faithfully,

The autors

This article assess the importance of intermediate respiratory care units (IRCU) as a support to ICU, in the context of the initial phase of the COVID-19 pandemic, when allocating care resources presented major difficulties.

This article presents IRCU as an alternative to alleviate high number of patients in ICU, and seemed to be associated with an overall reduced cost.

Although the objectives of the study are clearly defined and discussed throughout the study, I have some comments regarding data that are missing and other that could be better presented.

The text cites 2 tables (Table 1 and table 2) in the result section but they are not included while they could provide crucial additions.

According to the reviewer, I have attached the two tables that are amended in the text.

Table 1: Clinical variables and tests associated with mortality

Table 2: Hospitalization costs in the IRCU for a total of 56 patients over a two month period during the pandemic

'he NIRS which patients received upon admission to the IRCU': a typo is to be fixed in this sentence.

Thanks for the input from the reviewer, I'm changing the format of the chart to make it more understandable by following its directions.

Figure 2: could gain in clarity and it is difficult to understand. I suggest changing the fill colors or add a pattern to differentiate the survivors / deceased which appears the same color in black and white.  The 4 panes in this figure don't have a Y axis. I suggest adding 4 letters (A-D), one for each pane, and to indicate in the figure legend (A= fever; B= Therapy three grugs ... ) with additional details.

Thanks for the input from the reviewer, I'm changing the format of the chart to make it more understandable by following its directions.

Figure 3:  this figure is really poor quality and none of the text on both X and y axes is readable as well as the text on the right of the curves.

According to the reviewer, I modify the graphics, separating them to improve the quality of the image.

Reviewer 2 Report

It is my great pleasure to review this inspiratory manuscript. The introduction of a concept like  IRCUs (intermediate respiratory care units) in the era of non-invasive mechanical ventilation and high-flow oxygen therapy could be practical and beneficial! 

Minor points to be discussed:

In the part of Results, there is unclear patients flow and number and you had better reform Fig. 1.  For example, what is the meaning of the remaining six patients?

Sentences to expenditure need to be clarified. I do not understand that the average expenditure in the IRCU is higher than in the ICU.

There are some spelling errors to be corrected. 

Thank you a lot!

Author Response

It is my great pleasure to review this inspiratory manuscript. The introduction of a concept like  IRCUs (intermediate respiratory care units) in the era of non-invasive mechanical ventilation and high-flow oxygen therapy could be practical and beneficial! 

Minor points to be discussed:

In the part of Results, there is unclear patients flow and number and you had better reform Fig. 1.  For example, what is the meaning of the remaining six patients?

According to the reviewer, the paragraph is modified and the descriptive figure is changed, trying to minimize and facilitate the flow of patients in the different stages of severity.

Sentences to expenditure need to be clarified. I do not understand that the average expenditure in the IRCU is higher than in the ICU.

Please find an explanation in the discussion section. The fact that we have observed that the pharmaceutical expenditure is higher than the IRCU than in the individual ICU lies in the internal protocol itself and the follow-up of the scientific evidence of the moment. Drugs that in the case of NIRS did have evidence for use were not available in the patient under IoT. Thank you so much for your comment.

There are some spelling errors to be corrected. 

Thanks, these are now corrected. Thank you very much!

Reviewer 3 Report

Overall, the authors describe a prospective study on utilizing Intermediate Respiratory Care Units for COVID-19 patients as an alternative to the Intensive care units. While the authors describe significant cost-saving through their study, the methodology and results of the study do not align with the results and the content presented in the manuscript. Below are some of the recommendations to further improve the manuscript:

  1. The authors do not describe clearly how the IRCU defer from general ICU. While they do mention the staffing format of the IRCU, they fail to describe in detail how it is different from ICU and how were the IRCU created.
  2. Since this is a prospective study, the authors should have had a pre-defined selection criteria for admitting patients to the IRCU as oppose to the ICU. This is missing and needs to be clearly outlined. It is unclear if only patients on non-invasive respiratory support were admitted to IRCU or even the intubated patients were admitted.
  3. The authors describe predictors of mortality through regression analysis but it appears irrelevant to this study and primary aim of the study. The authors have to describe the primary and secondary endpoints/aims of the study clearly and present a statistical methodology appropriate for it. For example, it would be appropriate to have a descriptive table that presents demographics, lab and respiratory parameters of patients in IRCU in comparison to ICU patients and how they differ (using either parameteric or non-parametric test for means).

Otherwise, the authors describe appropriately their important findings of hospital days saving and cost saving per day which is interesting to the readership. However, the information gets lost and distracted with other results such as the regression analysis.

Author Response

Overall, the authors describe a prospective study on utilizing Intermediate Respiratory Care Units for COVID-19 patients as an alternative to the Intensive care units. While the authors describe significant cost-saving through their study, the methodology and results of the study do not align with the results and the content presented in the manuscript. Below are some of the recommendations to further improve the manuscript:

  1. The authors do not describe clearly how the IRCU defer from general ICU. While they do mention the staffing format of the IRCU, they fail to describe in detail how it is different from ICU and how were the IRCU created.

Thanks to the contribution, in the section of methods they modify the description of the unit as well as specific characteristics that differentiate it from the ICU.

  1. Since this is a prospective study, the authors should have had a pre-defined selection criteria for admitting patients to the IRCU as oppose to the ICU. This is missing and needs to be clearly outlined. It is unclear if only patients on non-invasive respiratory support were admitted to IRCU or even the intubated patients were admitted.

According to the reviewer, also in the methods section we define the admission criteria that we accepted during the first wave of the pandemic and detail that the patients with IoT needs were transferred to the ICU. Thank you so much for your comment.

  1. The authors describe predictors of mortality through regression analysis but it appears irrelevant to this study and primary aim of the study. The authors have to describe the primary and secondary endpoints/aims of the study clearly and present a statistical methodology appropriate for it. For example, it would be appropriate to have a descriptive table that presents demographics, lab and respiratory parameters of patients in IRCU in comparison to ICU patients and how they differ (using either parameteric or non-parametric test for means).

We appreciate the reviewer’s input,. Our goal is not to compare the ICU directly to the ICU, and therefore we cannot present clinical features of ICU patients and ICRI patients. The main goal and our starting point is mortality (patients who die and those who do not die) understanding that those patients who die were a priori more serious, but that they were not necessarily ICU patients (given the age and / or associated multiple pathologies), which is why we present the analyzes of the characteristics of the laboratory, clinics, etc. Depending on whether they die or not, and not depending on the resource (ICU, IRCU, hospitalization ...) they consume. It is important to understand that patients are not ‘pure’, in the sense that I am only in the ICU or UCRI, but that the same patient can go through different resources. To make it more definite, add a paragraph to the methods section

Otherwise, the authors describe appropriately their important findings of hospital days saving and cost saving per day which is interesting to the readership. However, the information gets lost and distracted with other results such as the regression analysis.

According to the reviewer, in the discussion section we have removed a paragraph in which a description of the characteristics of the patients as well as of the treatments received was made again. Thank you so much for your comment.

Reviewer 4 Report

The selected topic is excellent and generalized for several other audiences and researchers. The authors explain the alternative and effective method for ICU during the COVID_19 pandemic. However, I have some questions

whereas,

Is this curative activity, temporary or long-lasting?

 Is there any side-effects observed after the treatment?

Any significant mucus or BALF alterations found in patients?

Any specific mortality causing reasons are observed?

Any  physiological changes occurred  in the lung before and after the treatment? 

Author Response

The selected topic is excellent and generalized for several other audiences and researchers. The authors explain the alternative and effective method for ICU during the COVID_19 pandemic. However, I have some questions

whereas,

Is this curative activity, temporary or long-lasting?

Regarding the activity of the IRCU, it is of long duration (we refer to the section on methods that this unit already existed before the pandemic). These units have been very well received during the pandemic (although they were already used in many pulmonology services, they did not exist in all hospitals). With regard to the pharmacological treatments mentioned, it is true that they have varied progressively in the studies presented and are currently obsolete.

Is there any side-effects observed after the treatment?

Although we did not describe this in detail in the article, patients who underwent the pharmacological treatment described in our protocol had gastrointestinal disorders and minimal electrocardiographic disorders. In terms of noninvasive respiratory support, nasal septum ulcers secondary to the NMV interface were present in those patients with longer days of treatment; as well as occasional epistaxis in patients with long-term high-flow oxygen therapy. We add the input to the text in the results section.

Any significant mucus or BALF alterations found in patients?

During the study conducted in the article we present, we did not set the goal of describing specific sputum cultures, although we did collect the number of superinfections. We did not describe it in the article or analyze it in detail, but several patients who eventually needed IoT did have respiratory superinfections. We add the input to the text in the results section. Thank you so much for your comment.

Any specific mortality causing reasons are observed?

In the description of our results we determined that The mortality rate of the 81 patients admitted to the critical care unit was 25%. PaFiO2 at admission, PaO2, heart rate, RR, levels of lactate dehydrogenase (LDH), potassium, alanine aminotransferase (ALT) and total bilirubin were significantly associated with higher mortality in a univariate analysis. Thank you so much for your comment.

Any  physiological changes occurred  in the lung before and after the treatment? 

Thank you so much for your comment. During the study in the article we presented, we did not follow up the patients later (although they did in the hospital). We did not set the goal of describing the physiological changes made by non-invasive respiratory aids after passing COVID19. However, it would be a good proposal to consider in future studies.

Round 2

Reviewer 3 Report

The revised manuscript appears much improved. However, there are still significant concerns. 

  1. A descriptive table is important to highlight the characteristics of patients in IRCU for ease of readership. We are not expecting to compare them to ICU patients but more to understand how sick they are, what their demographics are and it serves as a good introductory table for the study.
  2. The authors decided to assess predictors of mortality, but have to detail their analysis methodology for it. It is not entirely clear if the result presented is just univariate analysis or multivariate analysis. 
  3. In general, the "methods" section needs significant revision as it is all lopped in one sub-section. Ideally, to make it easier for the readership and have clarity in the approach of your study, the Methods section could be sub-divided into several subcategories such as: 1. "Definitions": where you describe definitions of ICU, admission criteria etc. 2. "Inclusion and Exclusion Criteria": where you can describe criteria for including or excluding patients. 3. Statistical Analysis: This section is completely missing and needs more detailed explanation (e.g continuous variable described as mean or median, whether normality was tested, categorical as frequency/%, what analysis was performed, ?logistic regression etc).

Author Response

Dear reviewer,

Thank you very much for your comments, which have helped us very much to improve the text. Please find below our responses to your queries.

Yours faithfully,

The authors

------------------------

A descriptive table is important to highlight the characteristics of patients in IRCU for ease of readership. We are not expecting to compare them to ICU patients but more to understand how sick they are, what their demographics are and it serves as a good introductory table for the study.

-- According to the reviewer, we include a descriptive table of the characteristics of the patients admitted to the IRCU, taking into account the radiographic image, Bathel, Charlson index (associated comorbidities) and SOFA (severity) as well as arterial blood gas criteria at the beginning and at 3 days (TABLE 1). The gender of the patients was exactly 50% male and 50% female. The specifically associated comorbidities are detailed in the text.

The authors decided to assess predictors of mortality, but have to detail their analysis methodology for it. It is not entirely clear if the result presented is just univariate analysis or multivariate analysis.

-- Thank you very much for your comment. In accordance with the reviewer’s contribution, the paragraph in which the description of the type of analysis is made has been improved. We have now specified that we have performed an univariate analysis, whose main objective is to describe the data and find the patterns that exist in them.

In general, the "methods" section needs significant revision as it is all lopped in one sub-section. Ideally, to make it easier for the readership and have clarity in the approach of your study, the Methods section could be sub-divided into several subcategories such as: 1. "Definitions": where you describe definitions of ICU, admission criteria etc. 2. "Inclusion and Exclusion Criteria": where you can describe criteria for including or excluding patients. 3. Statistical Analysis: This section is completely missing and needs more detailed explanation (e.g continuous variable described as mean or median, whether normality was tested, categorical as frequency/%, what analysis was performed, ?logistic regression etc).

We have modified the methods section by adding the named subsections, as well as improving and increasing the description of the statistical analysis. Thank you very much for your comment.